# MULTI-BEHAVIOR DYNAMIC CONTRASTIVE LEARNING FOR RECOMMENDATION

## ABSTRACT

Dynamic behavior modeling has become an essential task in personalized recommender systems for learning the time-evolving user preference in online platforms. However, most next-item recommendation methods follow the single type behavior learning manner, which notably limits their user representation performance in reality, since the user-item relationships are often multi-typed in real-life applications (e.g., click, tag-as-favorite, review and purchase). To offer better recommendations, this work proposes Evolving Graph Contrastive Memory Network (EGCM) to model dynamic interaction heterogeneity for multi-behavior sequential recommendation. Specifically, we first develop a multi-behavior graph encoder to capture the short-term preference heterogeneity, and preserve the dedicated relation semantics for different types of user-item interactions. In addition, we design a dynamic cross-relational memory network, empowering EGCM to distill the long-term multi-behavior preference of users and the underlying evolving cross-type behavior dependencies over time. To obtain robust and informative user representation with multi-behavior commonality and diversity, we design a multi-behavior contrastive learning paradigm with heterogeneous short- and long-term interest modeling, and provides theoretical analyses to support the modeling of commonality and diversity. Experiments on several real-world datasets show the superiority of our recommender system over various state-of-the-art baselines.

## 1 INTRODUCTION

Learning user's dynamic preference plays a vital role in recommender systems to predict the next items that users may be interested in Wang et al. (2019). For example, a family may buy chicken and bread on an online platform for a long time because of their daily needs, and also buy turkeys close to Christmas. The recent advances of neural network architectures has inspired many efforts to model the transitions between temporally-ordered items, due to the strong representation capability of deep learning techniques, *e.g.*, recurrent neural encoder Hidasi et al. (2016), convolution-based model Tang & Wang (2018) and attention mechanism Kang & McAuley (2018). More recent sequential recommender systems are built upon the Transformer Sun et al. (2019); Liu et al. (2021b) or Graph Neural Networks (GNNs) Wu et al. (2019); Ma et al. (2020); Wang et al. (2020c) to provide state-of-the-art recommendation performance. Despite their effectiveness, most of existing next-item recommendation approaches rely on only single type of user-item interaction (*e.g.*, click or purchase data), and thus are limited to capture the item-level multi-behavior interaction patterns.

In real-life recommendation scenarios, users often interact with items in various ways, based on their interests which are intrinsically time-evolving and diverse. For instance, different types of user behaviors (*e.g.*, page view, add-to-favorite, purchase) in online retailers may reflect diverse user intentions and heterogeneous user-item relationships Guo et al. (2019); Jin et al. (2020b). Leaving this fact untouched, single type of behavior modeling in previous chronological user embedding functions is insufficient to comprehensively capture diverse user intents with behavior heterogeneity Xia et al. (2021a). Hence, time-evolving multi-behavior representations can characterize the various latent factors behind user-item interactions, and maintain dedicated embedding space for different types of dynamic user behaviors in recommender systems.

While having realized the importance of modeling behavior-aware time-evolving user-item relationships in recommendation, some key challenges remain to be carefully tackled. Specifically, (1) How to explicitly preserve the dynamic behavior-specific semantics pertinent to each type of user-item

interactions over time then delivering and retaining user preferences, is not trivial multi-behavior sequential recommendation. It is critical in for the recommender to distill such heterogeneous item-level dependencies with the jointly modeling of short-term and long-term user interests. (2) Learning informative and robust representations of multiplex user-item interactions requires a tailored modeling with a performant recommendation paradigm, which towards the encoding of users' multi-behavior commonality and diversity. While we can embed behavior-specific semantics into individual latent vectors, the understanding of multi-behavior commonality underlying global view of user-specific dynamic preference is critical to multi-behavior modeling.

**Contributions**. In light of these challenges, we propose an Evolving Graph Contrastive Memory (EGCM) framework that can effectively distill the heterogeneous user intentions over time from multi-behavior data in recommendation. Specifically, we first introduce a multi-behavior graph encoder equipped with temporal context embedding for modeling the behavior-aware short-term interests of users. Furthermore, a dynamic cross-relational memory network based on self-attention is proposed to incorporate heterogeneous cross-behavior dependencies into learning user dynamic preferences with cross-behavior relational transitions. In a nutshell, this dynamic multi-behavior modeling allows us to characterize diverse user intents from the long-term perspective behind the interacted item sequence. To enhance the generalizability and robustness of our recommender, we design our multi-behavior contrastive learning paradigm to endow EGCM with the capability of encoding multi-behavior commonality and differentiating the behavior-aware preference of various users. We also provide theoretical analysis of our EGCM model in Supplementary Section.

To summarize, the key contributions of this work are presented as follows:
- Emphasizing the importance of jointly learning of dynamic preference heterogeneity with multi-behavior data and diverse user behavior-aware interests for recommendation.

- Proposing a new model EGCM, which integrates the dynamic cross-relational dependency modeling with the multi-behavior contrastive learning paradigm, so as to distill the evolving user-item relationships at the fine-grained level of user preferences. In addition, we perform the theoretical analysis of our proposed EGCM model as presented in the Supplementary Section.

- Conducting experiments on three real-world datasets to demonstrate the superiority of EGCM. Further ablation studies and in-depth model analysis justify the rationality of our model design. To support the reproducibility of our experimental results, the model implementations can be found at the anonymous link: https://anonymous.4open.science/r/EGCM.

## 2 RELATED WORK

**Next-item/Sequential Recommendation**. Early studies (*e.g.*, FPMC Rendle et al. (2010)) rely on the Markov chain to tackle the sequential recommendation problem. Many recent efforts have been devoted to learning users' dynamic interests with various neural network encodes, such as the RNN-based method GRU4Rec Hidasi et al. (2016) and CNN-based approach Caser Tang & Wang (2018). In addition, several self-attention relational learning models are introduced to estimate the item correlations, *e.g.*, SASRec Kang & McAuley (2018), BERT4Rec Sun et al. (2019) and TiSASRec Li et al. (2020). Inspired by the strength of graph neural networks, some recent sequential recommender systems are built over the graph-based message passing scheme to encode the multi-order dependencies among items, including MA-GNN Ma et al. (2020), SURGE Chang et al. (2021), and GCE-GNN Wang et al. (2020c). Furthermore, self-supervised learning has been used in recent sequential recommendation methods for data augmentation, like COTREC Xia et al. (2021c) and DHCN Xia et al. (2021c). However, most of existing methods are built on single type of interactions and ignore the heterogeneous behavior-aware user preferences.

**Multi-Behavior Recommender System**. Recently, multi-behavior recommendation has gained considerable attention due to the effectiveness of considering multi-typed user behaviors in boosting the recommendation performance Chen et al. (2021). For example, NMTR Gao et al. (2019) and DIPN Guo et al. (2019) differentiating behavior semantics with multi-task learning schemes. To encode diverse relationships between users and items, some recent studies (*e.g.*, MBGCN Jin et al. (2020b), KHGT Xia et al. (2021a) and MBGMN Xia et al. (2021b)) attempt to leverage graph neural networks for encoding the multi-behavior patterns, based on their constructed relation-aware heterogeneous user-item interaction graph. One major drawback of existing multi-behavior recommender systems is that they mostly focus on the stationary scenarios, while neglect the time-evolving multi-behavior dependencies from diverse user interest representation.

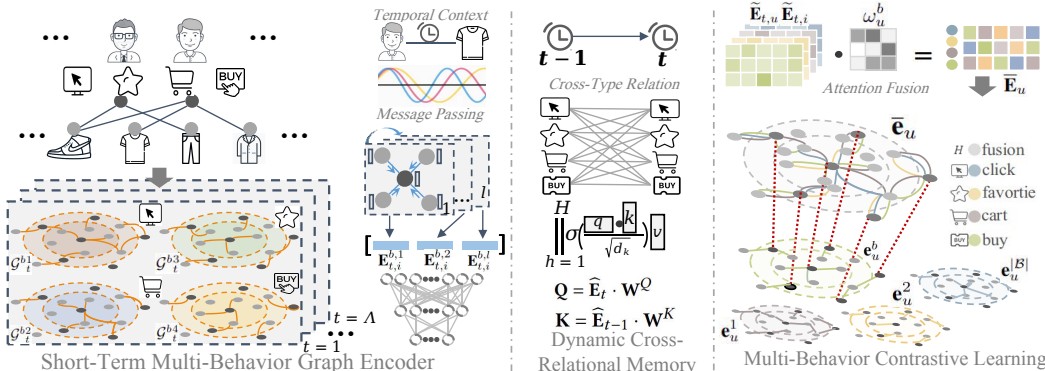

Figure 1: The model flow of EGCM framework. The multi-behavior contrastive learning module (right side) augments the graph-enhanced dynamic memory network (left & middle side) with the auxiliary behavior-aware self-supervision signals from both short-term and long-term user interests.

**Contrastive Representation Learning**. As a prevalent technique, contrastive learning has shown potential of auxiliary self-supervised signals Liu et al. (2021a); Ho & Nvasconcelos (2020) generation. For image analysis, many contrastive methods have been proposed to advance the modeling image data with different augmentation techniques Aberdam et al. (2021); Verma et al. (2021); Tian et al. (2020b), such as cropping, horizontal translations and rotations. In addition, cross-view contrastive learning has provided the state-of-the-art graph representation performance with various augmentation operators, *e.g.*, node shuffles in DGI Velickovic et al. (2019), centrality-aware edge dropout in GCA Zhu et al. (2021) and subgraph sampling in MVGLR Hassani & Khasahmadi (2020). Our work is inspired by the above work and designs a new multi-behavior contrastive learning paradigm to simultaneously capture the user-specific behavior commonalities and cross-user behavior diversity.

## 3 METHODOLOGY

We present the technical details of our EGCM model which consists of three key components: i) Short-term multi-behavior graph encoder that captures the user's short-term interests with multi-relational graph neural networks; ii) Long-term user interest modeling which learns time-evolving multi-behavior preferences across different time slots; iii) Multi-behavior self-supervised learning which enhances the user representation with the cross-behavior preference commonality and instance self-discrimination.

### 3.1 SHORT-TERM MULTI-BEHAVIOR GRAPH ENCODER

To capture the dedicated behavior semantics for underlying user's short-term interests, we propose a multi-relational graph encoder to handle the heterogeneous item dependencies within each time slot.

#### 3.1.1 SHORT-TERM MULTI-BEHAVIOR GRAPH

We define the short-term behavior-specific graph $\mathcal{G}_t^b = (\mathcal{V}_t^b, \mathcal{E}_t^b, \mathbf{M}_t^b)$ to represent the behavior-aware user-item interactions within the $t$-th time slot (*e.g.*, day, week, or month) under behavior $b$. And $\mathcal{G}_t^b$ is divided from the total interaction graph $\mathcal{G}$ by time-slot and behavior. In graph $\mathcal{G}_t^b$, nodes $\mathcal{V}_t^b = \mathcal{V}_{t,u}^b \cup \mathcal{V}_{t,i}^b$, and edges $\mathcal{E}_t^b$ represent the interactions for specific times and behaviors between user nodes $\mathcal{V}_{t,u}^b$ and item nodes $\mathcal{V}_{t,i}^b$. Given the set of edges $\mathcal{E}_t^b$, we define the user-item interaction matrix $\mathbf{M}_t^b$, where each entry $M_{t,(u,i)}^b = 1$ if user $u$ has adopted item $i$ under the behavior type of $b$ within the $t$-th time slot. $b \in \mathcal{B}$ denotes the specific user behavior(*e.g.*, click, tag-as-favorite, review and purchase). $t \in \Lambda$ denotes the current time slot.

#### 3.1.2 BEHAVIOR-AWARE MESSAGE PASSING

Inspired by recent efforts of utilizing graph neural network to model relational user data He et al. (2020); You et al. (2020), we design a behavior-aware message passing schema to capture the high-order dependencies among items through the recursively embedding propagation across graph layers. Formally, our behavior-aware message passing function from the ($l$-1)-th layer to the ($l$)-th layer is

defined with the following form:

$$\mathbf{E}_{t,i}^{b,(l)} = \sigma\left(\Gamma(\mathbf{M}_t^b \cdot \mathbf{M}_t^{b^T}) \cdot \mathbf{E}_{t,i}^{b,(l-1)} \cdot \mathbf{W}_{t,i}^{b,(l)}\right)$$

$$\Gamma(\mathbf{M}_t^b \cdot \mathbf{M}_t^{b^T}) = (\mathbf{D}_t^b)^{-\frac{1}{2}} \cdot \mathbf{M}_t^b \cdot (\mathbf{B}_t^b)^{-1} \cdot \mathbf{M}_t^{b^T} \cdot (\mathbf{D}_t^b)^{-\frac{1}{2}}$$

(1)

where $\mathbf{E}_{t,i}^{b,(l)}$ is the embedding of item $i$ at the $l$-th graph propagation layer at the $t$-th time slot under behavior $b$. $\sigma(\cdot)$ is the PReLU He et al. (2015) activation function to inject non-linearities. $\Gamma(\cdot)$ is the normalization function to alleviate the large value effects of embeddings during the recursive propagation Wang et al. (2020b). It is dedicated for the message passing on user-item heterogeneous bipartite graph, which is different from the symmetrical graph Laplacian of eigenvectors Kipf & Welling (2016). And the two diagonal degree matrices $\mathbf{D}_t^b \in \mathbb{R}^{N \times M}$ and $\mathbf{B}_t^b \in \mathbb{R}^{M \times N}$ are based on the interaction matrix $\mathbf{M}_t^b$. By stacking multiple embedding propagation layers, the behavior-specific high-order item dependencies can be preserved in the generated representations $\mathbf{E}_{t,i}^{b,(L)}$, where $L$ denotes the number of propagation layers. And we conduct the concatenation to aggregate the embeddings encoded from each graph layer, the concatenated representation is then sent into a linear layer with normalization $\mathbf{E}_{t,i}^b = \mathbf{E}_{t,i}^b / ||\mathbf{E}_{t,i}^b||$ to be reshaped to obtain output item representation $\mathbf{E}_{t,i}^b$ of out short term behavior-aware graph neural network.

After encoding the item embeddings by exploring the high-order dependencies among items, we generate the user representations which maintain the behavior-specific short-term interests:

$$\mathbf{E}_{t,u}^b = \sigma(\Gamma(\mathbf{M}_t^{b^T}) \cdot \mathbf{E}_{t,i}^b \cdot \mathbf{W}_{t,u}^b); \qquad \Gamma(\mathbf{M}_t^{b^T}) = (\mathbf{B}_t^b)^{-\frac{1}{2}} \cdot \mathbf{M}_t^{b^T} \cdot (\mathbf{D}_t^b)^{-\frac{1}{2}}$$

(2)

where $\mathbf{E}_{t,u}^b$ is corresponding to users' short-term interests at the $t$-th time slot under the behavior type of $b$ after normalization $\mathbf{E}_{t,u}^b = \mathbf{E}_{t,u}^b / ||\mathbf{E}_{t,u}^b||_2$. And the message passing from item to user is also over adjacency matrix $\mathbf{M}_{t,(u,i)}^b = 1/(\sqrt{|\mathcal{N}_u|}\sqrt{|\mathcal{N}_i|})$ with applying $\Gamma(\cdot)$.

### 3.1.3 PRIORI-AWARE INITIALIZATION & TEMPORAL CONTEXT INJECTION

In our short-term multi-behavior graph encoder, we rely on the time slot-specific user-item interaction data, which may lead to overfitting issue. In light of this limitation, we develop a priori-aware embedding initialization strategy which injects the representations of previous time slot $(t-1)$ into the embedding initialization of current time slot $t$, in order to reduce the training difficulty and deterioration Glorot & Bengio (2010). Formally, the priori-aware embedding initialization is defined as follows:

$$\mathbf{E}_{t,i}^b = \Upsilon(\mathbf{E}_{t,i,\mathbf{init}}^b) = \left(\zeta * \mathbf{E}_{t,i,\mathbf{init}}^b + (1-\zeta) * \mathbf{E}_{t-1,i}^b\right) \cdot \mathbf{W}_\zeta^b$$

(3)

where $\Upsilon(\cdot)$ is the initial function for preprocessing the Xavier initialied embedding $\mathbf{E}_{t,i,\mathbf{init}}^b$ for current behavior-specific short-term graph neural network which is apply before Eq. 1, Eq. **??**It's output $\mathbf{E}_{t,i}^b$ denotes the input embeddings for the short-term behavior-specific graph $\mathcal{G}_t^b$ at the $t$-th time slot. $\mathbf{E}_{t-1,i}^b$ represents the representations encoded from the previous $(t-1)$ time slot. Here, $\zeta$ and $\mathbf{W}_\zeta^b$ are learnable aggregation weight and projection matrix, respectively.

After representation learning, to inject temporal context signals into representation, our temporal context encoder is built over the positional encoding technique Vaswani et al. (2017). Specifically, the sinusoid functions was introduced to generate the relative time embedding for each user-item interaction edge. By doing so, we enhance our short-term multi-behavior graph encoder with the capability of capturing behavior-aware interaction dynamics of users with the temporal context injection.

### 3.2 DYNAMIC CROSS-RELATIONAL MEMORY NETWORK

In the multi-behavior recommender system, long-term multi-behavior dynamics reflect a holistic view of diverse preferences of users across different time slots. And different types of user behaviors are inter-dependent among time slots in a long-term perspective. For example, customers may first view some products in online retail sites, and make their purchase decisions one day after. To capture and convey such evolving cross-type behavior dependencies across time slots, we propose a dynamic cross-relational memory network to learn the evolving cross-type behavior dependencies over time in a long-term manner.

In order to form a memory network, we first learn the information to be remembered, *i.e.*, learn time-evolving cross-type behavior dependencies. Technically, it can be implemented by explicitly learning the influence weights between behavior-aware representations during specific time slot. Following the scaled dot-product self-attention function Vaswani et al. (2017), we take multi-behavior information at $t$ and $t-1$ as query($\mathbf{Q}$) and key($\mathbf{K}$), respectively. And value($\mathbf{V}$) is calculated from the embedding at $t-1$ to ultimately pass the information from $t-1$. Formally, our designed dynamic cross-relational memory encoder over adjacent time slot-specific behavior representations as follows:

$$\mathbf{Z}_{t,t-1} = \delta\left(\frac{\mathbf{Q}_t \cdot \mathbf{K}_{t-1}^T}{\sqrt{d/h}}\right) \cdot \mathbf{V}_{t-1}; \quad \begin{cases} \mathbf{Q}_t = \mathbf{H}_t \cdot \mathbf{W}_t^Q \\ \mathbf{K}_{t-1} = \mathbf{H}_{t-1} \cdot \mathbf{W}_t^K \\ \mathbf{V}_{t-1} = \mathbf{H}_{t-1} \end{cases} ; \quad \Phi = \mathbf{Q}_t \cdot \mathbf{K}_{t-1}^T \qquad (4)$$

where $\mathbf{Z}_{t,t-1}$ aggregates multi-behavior information of current time slot $t$ with the incorporation of cross-type behavior dependencies from the previous $(t-1)$-th time slot. $\mathbf{H}_{t,u}$ and $\mathbf{H}_{t,i}$ are three-way tensors to represent the multi-behavior short-term interest for each individual user prepare for self-attention module by stacking type-specific user($\mathbf{E}_{t,u}^b$) and item($\mathbf{E}_{t,i}^b$) embeddings. $\mathbf{W}_t^Q \in \mathbb{R}^{d\times d}$, $\mathbf{W}_t^K \in \mathbb{R}^{d\times d}$ represent the linear transformation matrices corresponding to the query and key dimension, respectively. The scale $\sqrt{d/h}$ is introduced to produce a softer attention distribution for avoiding gradient vanishing Vaswani et al. (2017), and $h$ represents the number of head representations. $\Phi$ is the learned self-attention weight matrix. Thus, each entry $\phi_{b,b'} \in \mathbb{R}^{|\mathcal{B}|\times|\mathcal{B}|}$ in $\Phi$ indicates the estimated interrelationship between the behaviors of adjacent time slot of a specific user. In our model implementation, we utilize the Broadcast Mechanism Van Der Walt et al. (2011) to improve the computational efficiency of tensor multiplication. Then, with the learned evolving cross-type behavior dependencies, we refine the time slot-specific user and item representations as:

$$\widetilde{\mathbf{E}}_{t,u} = \mu(\mathbf{E}_{t,u} \oplus \mathbf{Z}_{t,t-1,u}); \quad \widetilde{\mathbf{E}}_{t,i} = \mu(\mathbf{E}_{t,i} \oplus \mathbf{Z}_{t,t-1,i}) \qquad (5)$$

where $\widetilde{\mathbf{E}}_{t,u}, \widetilde{\mathbf{E}}_{t,i}$ are the embedding incorporate cross time information in terms of how various behaviors are related to one another. And $\mu(\cdot)$ indicates the adopted mean pooling operation. With the design of our dynamic cross-relational memory network, our EGCM can preserve the dedicated time-evolving behavior dependencies across different types of user interactions.

### 3.3 MULTI-BEHAVIOR CONTRASTIVE LEARNING

Having encoding the heterogeneous user interest representations corresponding to the short-term $(\mathbf{E}_{t,u}, \mathbf{E}_{t,i})$ and long-term $(\widetilde{\mathbf{E}}_{t,u}, \widetilde{\mathbf{E}}_{t,i})$ perspectives, we propose to jointly capture the diverse multi-behavior dependencies and commonality in the multi-behavior recommendation scenario.

#### 3.3.1 HETEROGENEOUS BEHAVIOR AGGREGATION

We first generate the multi-behavior representation $(\overline{\mathbf{E}}_u, \overline{\mathbf{E}}_i)$ and $(\overline{\widetilde{\mathbf{E}}}_{t,u}, \overline{\widetilde{\mathbf{E}}}_{t,i})$ by aggregating type-specific behavior representations with the attention mechanism for short term and long term, respectively. Take long-term representations as an example, it can be formally presented as follow:

$$\omega_u^b = \frac{exp(\widetilde{\mathbf{E}}_u^b \mathbf{W}_f)}{\sum_{b=1}^{|\mathcal{B}|} exp(\widetilde{\mathbf{E}}_u^b \mathbf{W}_f)}; \qquad \overline{\widetilde{\mathbf{E}}}_u = \sum_{b=1}^{|\mathcal{B}|} \omega_u^b \widetilde{\mathbf{E}}_u^b \qquad (6)$$

where $\omega_u^b$ represents the learned dependency weight of $b$-th type of user behaviors. And $\mathbf{W}_f \in \mathbb{R}^d$ is the transformation matrix before softmax. With the behavior aggregation attention mechanism, both short-term and long-term multi-behavior preferences of users can be preserved by the fused representations.

#### 3.3.2 CROSS-BEHAVIOR CONTRASTIVE SELF-SUPERVISION

In our EGCM model, we aim to simultaneously capture the multi-behavior commonality of individual user, and the multi-behavior distinction of different users. To this end, we design the cross-behavior contrastive learning module to enhance the multi-behavior dependency modeling with the augmented self-supervision signals. Specifically, we propose to generate our contrasting views with the type-specific behavior semantics and the fused multi-behaviour pattern. Our EGCM performs the behavior-level augmentation by pulling the type-specific behavior embedding ($\mathbf{e}_i^b \in \mathbb{R}^{d\times 1}$) and

Table 1: Statistics of experimented datasets

| Dataset | User # | Item # | Interaction # | Sparsity | Interaction Behavior Types |
|---|---|---|---|---|---|
| Taobao | 31882 | 31232 | 167862 | 99.98% | {View, Favorite, Cart, Purchase} |
| IJCAI | 22438 | 35573 | 199654 | 99.98% | {View, Favorite, Cart, Purchase} |
| E-commerce | 31021 | 1827 | 370386 | 99.35% | {Browse, Review, Purchase} |

multi-behavior representation ($\mathbf{e}_i \in \mathbb{R}^{d \times 1}$) of the same user $u$ closer as positive pairs, pushing the behavior embeddings of different users away as negative pairs. Formally, the cross-behavior contrastive objective is defined as:

$$\mathcal{L}_{cl}^b = \log \frac{exp(s(\mathbf{e}_u^b, \bar{\mathbf{e}}_u)/\tau)}{exp(s(\mathbf{e}_u^b, \bar{\mathbf{e}}_u)/\tau) + \sum_{u \neq u'} \left( exp(s(\mathbf{e}_u^b, \bar{\mathbf{e}}_{u'})/\tau) + exp(s(\mathbf{e}_u^b, \mathbf{e}_{u'}^b)/\tau) \right)} \quad (7)$$

where $\tau$ represents temperature parameter to control the effect of mutual information estimation Chen et al. (2020b). Here, we define the InfoNCE-based similarity function $s(\mathbf{e}_u^b, \bar{\mathbf{e}}_u) = \mathbf{e}_u^b \cdot \bar{\mathbf{e}}_u / \|\mathbf{e}_u^b\| \|\bar{\mathbf{e}}_u\|$ which is measured by the doc product between $\ell_2$ normalized $\mathbf{e}_u^b$ and $\bar{\mathbf{e}}_u$. With incorporating the cross-behavior contrastive self-supervision signals into our recommendation framework as auxiliary regularization, EGCMcan jointly capture the commonality of individual user intent and the diversity of different users with respect to their multi-behavior preferences. And, we talk about the details of commonality and diversity modeling in the supplementary material.

### 3.4 MODEL INFERENCE PHASE

For the main recommended task, we define our optimized objective with the Bayesian Personalized Ranking (BPR) loss as follows:

$$\mathcal{L}_{BPR} = \sum_{(u,i^+) \in \mathcal{E}^+, (u,i^-) \in \mathcal{E}^-} -\log(\text{sigmoid}(\bar{\mathbf{e}}_u \cdot \bar{\mathbf{e}}_{i+} - \bar{\mathbf{e}}_u \cdot \bar{\mathbf{e}}_{i-})) + \lambda\|\Theta\|^2 \quad (8)$$

where $\{(u, i^+, i^-)|(u, i^+) \in \mathcal{E}^+, (u, i^-) \in \mathcal{E}^-\}$ represents the pairwise training samples. Here, $\mathcal{E}^+$ and $\mathcal{E}^-$ denotes the corresponding observed and unobserved interaction of user $u$. To alleviate the overfitting issue, we apply the $L_2$ regularization in our BPR loss, and thus, $\Theta$ denotes the learnable hyperparameters. Finally, our joint optimization objective is given below by integrating BPR loss ($\mathcal{L}_{BPR}$) with short-term ($\sum_t^{|\Lambda|} \sum_b^{|\mathcal{B}|} \mathcal{L}_{cl}^{short}$) and long-term ($\sum_b^{|\mathcal{B}|} \mathcal{L}_{cl}^{long}$) contrastive objectives:

$$\mathcal{L} = \sum_b^{|\mathcal{B}|} \mathcal{L}_{BPR} + \alpha * \sum_b^{|\mathcal{B}|} \mathcal{L}_{cl}^{long} + \beta * \sum_t^{|\Lambda|} \sum_b^{|\mathcal{B}|} \mathcal{L}_{cl}^{short} \quad (9)$$

where $|\Lambda|$ represents the number of time slots. $\alpha$ and $\beta$ are regularization strengths of contrastive objectives. We further provide the in-depth analysis of our EGCM model with respect to the time complexity analysis and theoretical discussion in the submitted Supplementary material.

## 4 EVALUATION

### 4.1 EXPERIMENTAL SETUP

**Datasets**. We perform the model evaluation on three real-world datasets with the statistics shown in Table 1. i) **Taobao** is a benchmark dataset which collects four types of user behaviors from Taobao's recommender system. ii) **IJCAI** is a released online retailing dataset by IJCAI competition with four types of user online activities. iii) **E-commerce** contains browse, review and purchase behaviors of users in a real-life online retailer. Following the same settings in Jin et al. (2020a); Xia et al. (2021b), our multi-behavior sequential recommender system regard the purchase as the target behaviors and other types of user behaviors as auxiliary behaviors.

**Evaluation Protocols**. In our evaluation, two representative metrics, *i.e.*, Hit Ratio (HR@N) and Normalized Discounted Cumulative Gain (NDCG@N) ($N = 10$ by default), are adopted to measure the recommendation accuracy. To be consistent with the settings in existing sequential recommender systems Sun et al. (2019), the leave-one-out evaluation strategy is used for evaluation. Additionally,

for each user, the last interacted item under the target behavior type is considered as the positive samples in the test set and 99 non-interacted items are randomly sampled as negative instances.

**Baselines**. We compare our EGCM with various types of state-of-the-art recommendation methods.

- (i) CNN/Attention-based Sequential Recommendation Approaches: **Caser** Tang & Wang (2018) uses the convolutional filters to encode local item dependencies of user sequences. **SASRec** Kang & McAuley (2018), **TiSASRec** Li et al. (2020), **AttRec** Zhang et al. (2018) are built on self-attention mechanism to capture correlations between temporally-ordered items. **Bert4Rec** Sun et al. (2019) trains the bidirectional Transformer model with the cloze task.
- (ii) Hybrid Sequential Recommender Systems: **HGN** Ma et al. (2019) is a hierarchical gating network to learn the item feature relevance. **Chorus** Wang et al. (2020a) considers both knowledge-aware item relations and temporal context in sequential recommendation.
- (iii) GNN-based Sequential Recommendation Models: **SR-GNN** Wu et al. (2019) first introduces the graph neural networks to model the sequential behavior patterns over short item sequences. **MA-GNN** Ma et al. (2020) leverages GNNs to encode both short-term and long-term item dependencies. Furthermore, **HyperRec** Wang et al. (2020b) proposes to capture dynamic triadic item relationships using the hypergraph structures. **COTREC** Xia et al. (2021c) and **DHCN** Xia et al. (2021d) are two state-of-the-art sequential recommender systems based on self-supervised learning paradigms.
- (iv) Multi-Behavior Recommender Systems: **NMTR** Gao et al. (2019) and **DIPN** Guo et al. (2019) formalize the multi-behavior recommendation task with multi-task learning frameworks. **MBGCN** Jin et al. (2020b), **KHGT** Xia et al. (2021a) and **MBGMN** Xia et al. (2021b) design multi-behavior graph message passing schemes to model heterogeneous user-item interactions. **EHCF** Chen et al. (2020a) and **CML** Wei et al. (2022) generate additional supervision signals from auxiliary behaviors to boost the recommendation performance.

Table 2: Performance comparison of all methods on different datasets in terms of *HR & NDCG*.

|  | Caser | Att Rec | SAS Rec | TiSAS Rec | BERT 4Rec | HGN | Chorus | SR- GNN | MA- GNN | Hyper Rec | DHCN |
|---|---|---|---|---|---|---|---|---|---|---|---|
| Tmall | | | | | | | | | | | |
| H@10 | 0.321 | 0.328 | 0.319 | 0.322 | 0.329 | 0.283 | 0.335 | 0.318 | 0.331 | 0.333 | 0.321 |
| N@10 | 0.195 | 0.197 | 0.184 | 0.185 | 0.197 | 0.172 | 0.201 | 0.189 | 0.202 | 0.204 | 0.193 |
| IJCAI | | | | | | | | | | | |
| H@10 | 0.257 | 0.261 | 0.263 | 0.262 | 0.281 | 0.251 | 0.270 | 0.280 | 0.259 | 0.265 | 0.271 |
| N@10 | 0.146 | 0.147 | 0.148 | 0.148 | 0.155 | 0.136 | 0.149 | 0.151 | 0.149 | 0.151 | 0.148 |
| E-commerce | | | | | | | | | | | |
| H@10 | 0.627 | 0.628 | 0.599 | 0.599 | 0.641 | 0.586 | 0.639 | 0.620 | 0.644 | 0.648 | 0.638 |
| N@10 | 0.381 | 0.387 | 0.368 | 0.368 | 0.392 | 0.359 | 0.402 | 0.373 | 0.411 | 0.413 | 0.391 |

|  | COTREC | NMTR | DIPN | MBGCN | KHGT | MBGMN | EHCF | CML | **EGCM** | Imprv. | p-value |
|---|---|---|---|---|---|---|---|---|---|---|---|
| Tmall | | | | | | | | | | | |
| H@10 | 0.330 | 0.362 | 0.325 | 0.381 | 0.391 | 0.419 | 0.433 | 0.543 | **0.597** | 9.94% | $5.2e^{-5}$ |
| N@10 | 0.201 | 0.215 | 0.193 | 0.213 | 0.232 | 0.246 | 0.260 | 0.327 | **0.363** | 11.01% | $4.9e^{-6}$ |
| IJCAI | | | | | | | | | | | |
| H@10 | 0.278 | 0.269 | 0.276 | 0.270 | 0.278 | 0.329 | 0.362 | 0.410 | **0.510** | 24.39% | $9.6e^{-6}$ |
| N@10 | 0.153 | 0.156 | 0.151 | 0.138 | 0.145 | 0.176 | 0.207 | 0.235 | **0.312** | 32.77% | $2.3e^{-5}$ |
| E-commerce | | | | | | | | | | | |
| H@10 | 0.647 | 0.651 | 0.655 | 0.679 | 0.689 | 0.690 | 0.611 | 0.719 | **0.763** | 6.12% | $8.7e^{-3}$ |
| N@10 | 0.405 | 0.408 | 0.397 | 0.414 | 0.434 | 0.432 | 0.413 | 0.427 | **0.470** | 10.07% | $3.4e^{-4}$ |

\* To save space, we folded the table into two parts. Where the bolded columns are our results, followed by the *improvment* and *p-value* for the best results.

**Hyperparameter Settings and Implementation Details**. We implement our model in Pytorch and adopt Xavier initializer Glorot & Bengio (2010) for parameter initialization. The Cyclical Learning Rate strategy Smith (2017) is used during the model training phase with the AdamW Loshchilov & Hutter (2017) optimizer. For fair comparison, the number of graph layers in all GNN-based models are selected from the range {1,2,3,4} to achieve the best performance. The temperature coefficient $\tau$ is tuned from {0.02, 0.035, 0.05, 0.07, 0.1, 0.3, 0.5, 0.7} in our multi-behavior contrastive learning component. We further study the influence of key hyperparameters in our model.

## 4.2 RECOMMENDATION PERFORMANCE

We report the performance comparison results in Table 2 and summarize the findings: (1) EGCM outperforms various baselines in all cases by achieving significant performance improvements. The "imprv" indicates the relative performance improvement between EGCM and the best performed baseline CML. Through the encoding of the evolving dependencies across different types of behaviors, EGCM is able to simultaneously capture the dynamics of users' short-term and long-term interests, by distilling the underlying heterogeneous interaction patterns. (2) EGCM outperforms the compared sequential recommender systems by a large margin, which indicates that the incorporation of multi-behavior context is beneficial for disentangling behavior heterogeneity of users. (3) Conducting dynamic contrastive learning with cross-behavior dependencies, EGCM learns better multi-behavior representations compared with state-of-the-art multi-behavior recommendation approaches, by preserving both the behavior commonality and diversity of users. (4) The consistent performance improvements of our method on datasets with different sparsity degrees, benefits from the incorporation of effective self-supervision signals generated by our contrastive learning paradigm.

## 4.3 ABLATION STUDY

We evaluate the efficacy of key components in EGCM with different variants: (1) *w/o-JBL*: Instead of performing jointly learning with multi-typed behavior supervision labels, we directly incorporate auxiliary behavior data as contextual features for user representation. (2) *w/o-CL*: We further disable our multi-behavior contrastive learning component to capture the behavior commonality and diversity. (3) *r/w-GRU*: We replace our dynamic cross-relational memory network with gated recurrent unit (GRU) to encode behavior-specific embedding sequences. (4) *w/o-MBG*: On the basis of *w/o-CL*, we remove the behavior-aware graph neural encoder, to model the high-order connectivity over the multi-behavior user-item interaction graph within a specific time slot.

From evaluation results shown in Table 3, we can observe that our multi-behavior learning with augmented short- and long-term contrastive objectives improves the generalization of multi-behavior recommender system. In addition, our cross-relational memory network is more effective than GRU to encode long-term multi-behavior user interests. Furthermore, the multi-behavior graph encoder brings positive effects to learn short-term heterogeneous user preferences, compared with directly information aggregation over id-corresponding embeddings for collaborative effect modeling.

## 4.4 IN-DEPTH ANALYSIS OF EGCM MODEL

**Commonality and diversity.** For cross-behavior user preference commonality, to observe the effect of mutual information maximization, in Figure 2 (a)-(d), we visualize the encoded behavior-specific user embeddings of our EGCM and the variant w/o-CL (without the multi-behavior contrastive learning) on Tmall and IJCAI, respectively. Theoretically, limiting contrastive loss is maximizing the lower bound for mutual information and the rationality of multi-view contrastive learning(Supplementary materials A.1.1), which support this operation. The visualization suggest that our cross-behavior multi-view contrastive paradigm is effective to preserve user's multi-behavior commonality by bringing the multiple behaviors semantics close to each other, rather than keep behavioral representations in their own space. For user representation diversity, as detailed in the Supplementary Material.A.1.2, the gradient of the negative sample sample of the anchor(each user $u$) is proportional related as follow equation:

$$c(x) \propto \sqrt{1 - (x)^2} \cdot \exp\left(x/\tau\right) \qquad (10)$$

where $c(x)$ is the relationship function of the gradient from the negative samples, $x$ represents the similarity between the anchor node and the paired sample(*e.g.*, $x = \mathbf{e}_u^b, \bar{\mathbf{e}}_u \in [-1, 1]$). And we plot $c(x)$ in right part of Figure 2 (e) when temperature coefficient $\tau = \{0.02, 0.03, 0.2\}$. It can be observed that the gradient appears to grow exponentially, when $\tau$ decrease while similarity $x$ increase gradually. Explained from the perspective of the recommendation task, anchor node indicates specific user, and negative samples represent other $|\mathcal{V}_u| - 1$ users. If other users are close to the anchor user, i.e., the over-smoothing may have occurred and result in indistinguishable user representations. In this case, the contrastive loss will give a larger gradient to those users to pushed them away. Therefore, with cross-behavior contrastive self-supervision, EGCM can better differentiate diverse user preference and enhance the user representations with multi-behavior diversity. Therefore, it can be seen from left part of Figure 2 (e) that as the temperature $\tau$ decreases which, the more distinguishable the user representation is, which brings the better effect. However, when the gradient is quite large, the 'NaN' value which due to gradient explosion will be observed. as shown in Figure 2

Table 3: Ablation study on the effectiveness of components in EGCM.

| Data | Tmall | | IJCAI | | E-commerce | |
|------|-------|---|-------|---|------------|---|
| Metrics | HR@10 | NDCG@10 | HR@10 | NDCG@10 | HR@10 | NDCG@10 |
| $w/o$-MBG | 0.338 | 0.219 | 0.261 | 0.139 | 0.630 | 0.372 |
| $r/w$-GRU | 0.361 | 0.210 | 0.276 | 0.148 | 0.675 | 0.417 |
| $w/o$-CL | 0.391 | 0.231 | 0.300 | 0.166 | 0.734 | 0.461 |
| $w/o$-JBL | 0.473 | 0.291 | 0.367 | 0.205 | 0.747 | 0.469 |
| EGCM | **0.597** | **0.363** | **0.510** | **0.312** | **0.763** | **0.470** |

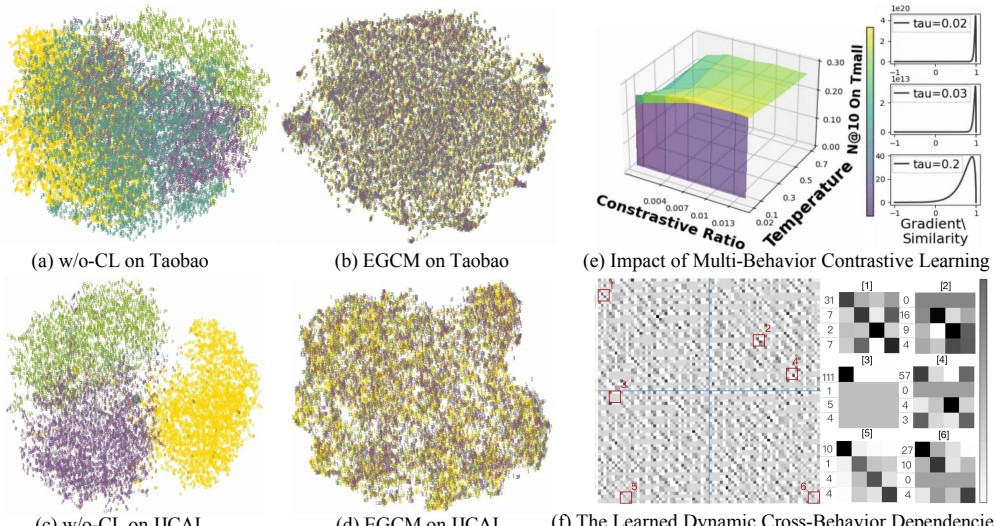

(a) w/o-CL on Taobao    (b) EGCM on Taobao    (e) Impact of Multi-Behavior Contrastive Learning

(c) w/o-CL on IJCAI    (d) EGCM on IJCAI    (f) The Learned Dynamic Cross-Behavior Dependencies

Figure 2: (a)-(b): Visualized behavior-aware user representations which preserve multi-behavior commonality and cross-user behavior diversity. (e): Impact study of contrastive learning in terms of temperature parameter $\tau$(see Sec. A.1.2 for details). (f): Learned cross-behavior dependency weights across time slots.

(e), we can notice that a smaller value of temperature value $\tau$ may bring larger gradients for better identifying hard negatives, so as to enhance the model discrimination ability in learning personalized user interests.

**Case study of memory network.** iii) In Figure 2 (f), we visualize the learned cross-behavior dependencies between time slots with $4 \times 4$ weight matrices $\phi \in \mathbb{R}^{|\mathcal{B}| \times |\mathcal{B}|}$ for 400 sampled users on Taobao dataset(with Page View, Favorite, Cart, Purchase four kinds behaviors). Then, we select six cases from the sampled data and combine the real statistics of multi-behavior data to analyze the significance of the learned weight matrices. It can be observed in Figure 2 (f) that the weight of each behavior is related to the number of interactions of the behavior. For example, for user 22186 in Figure 2[3] with 111 interactions in *page view* behavior, the number of interactions in other behaviors are $\{1, 5, 4\}$ which are too small relatively. Thus it is difficult to learn differentiated values in the other three rows relative to the first row of the weight matrix. In addition, it can be seen from the thumbnail, most matrices have the darkest diagonal color, which is the characteristic of self-attention.

## 5 CONCLUSION

This work studies the problem of multi-behavior sequential recommendation and propose our model that captures the interaction heterogeneity of each user at both the short-term and long-term level interests. In addition, a multi-behavior contrastive learning paradigm is presented to not only model the multi-behavior commonalities of individual user, but also enhances the distinction of behavior-aware preference of different users. Experiments on three real-world datasets demonstrate that EGCM significantly advances the recommendation performance compared with many strong baselines. In the future, we plan to extend EGCM to adapt to the cross-domain recommendation to tackle the cold-start problem in multi-behavior recommender systems.

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
