# OpenReview forum: "Multi-Behavior Dynamic Contrastive Learning for Recommendation"
_ICLR.cc/2023/Conference — Submitted to ICLR 2023_

### Official Review · Reviewer_HypT · 2022-10-24

**Confidence:** 4
**Correctness:** 4
**Technical Novelty And Significance:** 3
**Empirical Novelty And Significance:** 4
**Recommendation:** 10

**Clarity, Quality, Novelty And Reproducibility:**

This paper solves an under-explored recommendation problem with dynamic diverse behavior patterns. A novel contrastive learning paradigm is designed to tackle the corresponding technical challenges. Sufficient and appropriate experiments are conducted to show the new recommender EGCM’s effectiveness. The source code and evaluation data are released for result reproducibility. The parameter settings are described in the paper.

**Strength And Weaknesses:**

Pros:
1.The manuscript is well-organized and easy to follow.
2. The descriptions of model motivation with technical details are very clear. The authors point out key challenges in learning preference dynamics for multi-behavior recommender systems. In the proposed EGCM method, the multi-behavior dynamic dependency challenges are addressed by integrating the contrastive SSL with cross-relation memory network.
2.Theoretical discussion of the new contrastive learning is offered to analyze the benefits of the introduced behavior-wise contrastive-based SSL.
3.Comprehensive experiments are conducted to justify the effectiveness of the new approach by comparing EGCM with 19 alternative solutions.
4.Component-wise model evaluation as well as the in-depth analysis of EGCM method are provided to show the rationale of the superior performance.

Cons:
In the model inference phase, the self-supervision loss objectives L_{cl} with both short-term and long-term behavior modeling are incorporated into the original supervision optimized BPR-based loss. Besides the provided pseudo code in the appendix, the details of learning process in EGCM can be elaborated.

For the experiment of alleviating data sparsity issue, while the variants already achieve the obvious performance gain, the recommendation results of the new method can also be added.


**Summary Of The Paper:**

This paper studies the problem of capturing the interaction heterogeneity in modeling dynamic user preference for recommendation. A new evolving graph contrastive learning paradigm is designed to preserve both multi-behavior diversity and commonality. Extensive experiments are conducted to demonstrate the effectiveness of the new method compared with a large number of alternative solutions from different perspectives.

**Summary Of The Review:**

The proposed new framework would be beneficial for contrastive learning domain in learning the relational dynamics. The idea of dynamic contrastive learning to behavior-aware recommendation is novel and interesting. Good performance with comprehensive experiments on benchmark datasets are given.

---

> ### Author Response · Authors · 2022-11-19
> **Modifications Made During Rebuttal**
>
>
> We are quite grateful that the reviewer thoroughly read our and and noted the positive aspects.
> Ans thank you so much for your acknowledgement in our model technical novelty and comprehensive experiments.
>
>
> * The methods section has been rewritten to make it easy for readers to follow.
> * Detailing the Sec.4.4: In-depth Analysis Of Egcm Model.
> * The experiment of ablation behavior and all item test is supplemented.
> * Added dimensional transformation table(Appendix.A.7) for the memory module to illustrate the difference from the normal transformer.
> * Added descriptions and details of experiments that have been done(Appendix.A.9).

---

### Official Review · Reviewer_1ytF · 2022-10-24

**Confidence:** 3
**Correctness:** 3
**Technical Novelty And Significance:** 3
**Empirical Novelty And Significance:** 2
**Recommendation:** 6

**Clarity, Quality, Novelty And Reproducibility:**

The clarity of the paper should be further improved.

The overall quality looks good.

Novelty is limited since many modules are simple applications/extensions of existing methods.

Reproducibility is good. The implementation details are presented and the code is also provided.

**Strength And Weaknesses:**

Strengths:

1. The experimental results show that the proposed method could significantly outperform the SOTA baselines.
2. Code is provided.

Weaknesses:
1. The novelty of the paper is limited. Many modules are simple applications/extensions of existing methods.
2. The proposed model is too complex.
3. The writing is not quite clear, and some parts are difficult to follow.

For example, Eq. (4) is difficult to understand. Where to use $\mathbf{m}^b_{u_t\leftarrow i_t}$?

You use $f({\mathbf{e}^b_{i_t}}, {\mathbf{e}^b_{u_t}})$ in Eq. (4), however, ${\mathbf{e}^b_{u_t}}$ does not appear on the right hand side.

**Summary Of The Paper:**

This paper introduces an Evolving Graph Contrastive Memory (EGCM) framework which effectively models the multiple behaviors in the recommendation. EGCM is comprised of (1) a multi-behavior graph encoder to model the behavior-aware short-term interests of users, (2) a dynamic cross-relational memory network to model cross-behavior relational transitions, (3) a contrastive learning module to enhance the generalizability and robustness. The experimental results show that the proposed method could effectively outperform SOTA methods.

**Summary Of The Review:**

The experimental results demonstrate that the proposed method could significantly outperform SOTA methods. However, the novelty is limited.

---

> ### Author Response · Authors · 2022-11-19
> **Modifications Made to the Article's "method" Section**
>
> We spent a lot of effort organizing and updating the "methods" part in response to the second and third comments highlighted by the reviewer in the "Weaknesses" section. We believed that these issues were related to the poor writing in that section.
>
> * After the reorganization, the number of formulas in Sec. 3.1 part has been reduced by 3 and the number of formulas in Sec. 3.2 part has been reduced by 1. If our revised version can be easily followed by readers and approved by the reviewer, we would be overjoyed.

---

> > ### Comment · Reviewer_1ytF · 2022-12-07
> > **Response**
> >
> > The authors have addressed my concerns and I'd like to raise my score.

---

> > > ### Author Response · Authors · 2022-12-07
> > > **We appreciate your acknowledging our work.**
> > >
> > > We want to express our sincerely gratitude for your time to review our work and give us constructive comments. Taking into account your comments, we have revised the article, which has helped a lot to improve the content.

---

> ### Author Response · Authors · 2022-11-19
> **Differences from Previous Work(Part2)**
>
>
>
> | Dimensional Transformation of the Memory Module |                                                                                                                                                                                                                          |
> |:-----------------------------------------------:|--------------------------------------------------------------------------------------------------------------------------------------------------------------------------------------------------------------------------|
> | Parameters                                      | Dimensionality                                                                                                                                                                                                           |
> | Input                                           | ($\|\mathcal{B}\|$ $\times$ N $\times$ d)                                                                                                                                                                                |
> | Q,K,V Transformation                            | ($\|\mathcal{B}\|$ $\times$ N $\times$ d) $\cdot$ (d $\times$ d) $\longrightarrow$  ($\|\mathcal{B}\|$ $\times$ N $\times$ d)                                                                                            |
> | Q Extension                                     | ($\|\mathcal{B}\|$ $\times$ N $\times$ d) $\longrightarrow$  ($\|\mathcal{B}\|$ $\times$ 1 $\times$ N $\times$ d)                                                                                                        |
> | K Extension                                     | ($\|\mathcal{B}\|$ $\times$ N $\times$ d) $\longrightarrow$ (1 $\times$ $\|\mathcal{B}\|$ $\times$ N $\times$ d)                                                                                                         |
> | V Extension                                     | ($\|\mathcal{B}\|$ $\times$ N $\times$ d) $\longrightarrow$ (1 $\times$ $\|\mathcal{B}\|$ $\times$ N $\times$ d)                                                                                                         |
> | Self-attention                                  | ($\|\mathcal{B}\|$ $\times$ 1 $\times$ N $\times$ d) $\cdot$ (1 $\times$ $\|\mathcal{B}\|$ $\times$ N $\times$ d)  $\longrightarrow$   ($\|\mathcal{B}\|$ $\times$ $\|\mathcal{B}\|$ $\times$ N $\times$ d)              |
> | Reduce Sum                                      | ($\|\mathcal{B}\|$ $\times$ $\|\mathcal{B}\|$ $\times$ N $\times$ d)       $\longrightarrow$   ($\|\mathcal{B}\|$ $\times$ $\|\mathcal{B}\|$ $\times$ N $\times$ 1)                                                      |
> | Softmax                                         | ($\|\mathcal{B}\|$ $\times$ $\|\mathcal{B}\|$(softmax) $\times$ N $\times$ 1)                                                                                                                                            |
> | Attention Matrix*V                              | ($\|\mathcal{B}\|$ $\times$ $\|\mathcal{B}\|$ $\times$ N $\times$ d) $\cdot$ (1 $\times$ $\|\mathcal{B}\|$ $\times$ N $\times$ d) $\longrightarrow$ ($\|\mathcal{B}\|$ $\times$ $\|\mathcal{B}\|$ $\times$ N $\times$ d) |
> | Output                                          | ($\|\mathcal{B}\|$ $\times$ $\|\mathcal{B}\|$(reduce sum+squeeze) $\times$ N $\times$ d) $\longrightarrow$ ($\|\mathcal{B}\|$ $\times$ N $\times$ d)                                                                     |

---

> ### Author Response · Authors · 2022-11-19
> **Differences from Previous Work(Part1)**
>
>
> First of all, we appreciate reviewer for raising relevant doubts and please allow us to give an explanation:
> * Our module does apply classical models and paradigms, which contribute to better experimental results. However, we are considering how to model long- and short-term multi-behavior paradigms in user-item interactions with rare validated classical backbone models(e.g., Diffusion Model, GNN, Contrastive and Transformer).
>
> * Each module of our model is designed for scenario-specific data, they are not simple applications of the backbone model. We did many substitution experiments during the design process and used the version that made the best results in the end. In addition, we performed theoretical support(e.g., contrast learning for multi-view mutual information maximization in *Appendix.A.1.1* and increased instance differentiation by increasing the gradient in *Appendix.A.1.2*) for some modules and obtained experimental results that echoed the theory(The theory in A.1.1 can explain the phenomenon in *Sec.4.4.(e)*, i.e. gradient explosion).
>
> * Allow us to present the advantages of our modules over the backbone model:
>     * Transformer：
>         * We do not use normal transformer in the model, but we introduce self-attention to model the relationship of multiple behaviors between time slot t and t-1.
>         * To illustrate that our self-attention is used to model the relationship between each user's behavior rather than the tranfomer component, we have added the process of tensor dimension transformation table(*Appendix.A.7*) and expanded the description(*Appendix.A.9*) of the self-attention weight visualization experiment to the supplementary material.
>
>     * Contrastive：
>         * Our contrastive module models the commonality and diversity of user preference. And we provide theoretical support(*Appendix.A.1*) and experimental validation(*Sec.4.4.(f)*) for "commonality" and "diversity" respectively. What's more, it is worth noting that, some experimental results with parameters not in the reasonable range are the extreme cases of the theoretical analysis.
>         * Commonality: We believe that there are user common preferences among multiple behaviors. By performing mutual information maximization across behaviors, more accurate and robust user representations can be obtained. The theoretical analysis section also states that minimizing our multibehavioral cl loss is approximately a lower bound(*Eq.11*) for maximizing multibehavioral mutual information. In addition, the theoretical analysis section provides us with a theoretical analysis(*Eq.12*) for multi-view contrastive learning.
>         * Diversity: The modeling of short-term user-item interactions is based on GNN, which helps to model higher-order connectivity[1]. However, the multilayer GNN architecture introduces over-smoothing problems. The theoretical analysis part points out that the cl loss gives a larger gradient to the difficult negative samples(*Appendix.A.1.2.Eq.18*), which means push away similar user representations, i.e., mitigating the over-soomthing problem. And, very interestingly, our experiments(*Sec.4.4*) also corroborate with the theoretical analysis. That is, a small temperature coefficient does give a larger gradient, but too large may lead to a gradient explosion. We publish the code, data set, and parameters of the model (e.g., a Tmall temperature coefficient of 0.037 is optimal, but a loss of "NaN" occurs at 0.02 because of gradient explosion).
>
>
> > [1] Wang, Xiang, et al. "Neural graph collaborative filtering." Proceedings of the 42nd international ACM SIGIR conference on Research and development in Information Retrieval. 2019.

---

### Official Review · Reviewer_rXXA · 2022-10-24

**Confidence:** 4
**Correctness:** 2
**Technical Novelty And Significance:** 2
**Empirical Novelty And Significance:** 2
**Recommendation:** 6

**Clarity, Quality, Novelty And Reproducibility:**

Despite the ideas being relatively straight-forward, I found the paper quite difficult to read. I found the mathematical presentation particularly poor. In particular, almost everything is an E with various subscripts, superscripts, and accents. Many terms are not defined after they are first used, and the dimension of numerous variables is not explicitly.

Given the large number of architectural components, I also suspect it would be difficult to reproduce the results in the paper, and the authors do not provide anonymized source code. As this has been a persistent issue in the space [3], I would strongly encourage the authors to consider whether their results would be reproducible.

**Strength And Weaknesses:**

**Strengths**

- The method performs well in the experiments.
- Some ablation studies are included to validate choices.

**Weaknesses**

- The constituent pieces of the proposed model are known and there is limited technical novelty.
- Many of the design choices seem arbitrary and are not adequately justified. Over three pages of the paper are spent describing the architecture and providing fairly loose intuitive motivation without actually justifying why the specific choice is obvious or should be preferred, e.g., "both short-term and long-term multi-behavior preferences of users can be preserved by the fused representations," "our EGCM can preserve the dedicated time-evolving behavior dependencies across different types of user interactions." The desirability of these properties may be intuitive, but how this maps to the specific choices made is not.
- The ablation studies are extremely limited, and the experiments are not designed to justify the design decisions made. For example, without designing experiments that explicitly test the model's ability to "capture such evolving cross-type behavior dependencies across time slots"  how can the reader be sure this is what the proposed mechanism is actually doing? Indeed, in the results section the authors somewhat tautologically explain the observed performance improvements, "EGCM outperforms various baselines in all cases by achieving significant performance improvements." The lack of rigorous experimental design to validate the reason for performance improvements makes me doubtful that the paper provides a solid foundation on which subsequent research can build.
- The paper cites [1], although it does not seem to take any lessons from the work. For example, the arguments that applying non-linearities and transformation matrices to non-feature-based representations (e.g. a user or item id) seem to apply to this work as well.
- The authors use sampled evaluation metrics despite previous work [2] showing this methodology can be misleading. Since the datasets seem relatively small (the largest dataset features 35.5K items), I'm curious if there was a particular reason the paper chose this evaluation.
- The "In-Depth Analysis of ECGM Model" section is only a paragraph, and the conclusions drawn from the figures are not adequately described. I didn't find this section particularly interesting, informative, or in-depth.

**References**
1. [LightGCN: Simplifying and Powering Graph Convolution Network for Recommendation](https://arxiv.org/abs/2002.02126)
2. [On sampled metrics for item recommendation](https://dl.acm.org/doi/abs/10.1145/3535335)
3. [A Troubling Analysis of Reproducibility and Progress in Recommender Systems Research](https://dl.acm.org/doi/abs/10.1145/3434185)

**Summary Of The Paper:**

This paper proposes a graph neural network (GNN) based recommender system for datasets featuring heterogeneous interactions (click, purchase, etc). This is done by performing message passing within behavior and time slot-specific subgraphs. Information is fused between behavior types and across time slots using various standard neural network components. Additionally, the paper proposes to use contrastive learning to encourage representations for the same user to be similar while pushing apart representations for different users. Experiments demonstrate the proposed model outperforms baselines on two small benchmark datasets and a proprietary (I think) e-commerce dataset.

**Summary Of The Review:**

While the experiments demonstrate performance improvements over baselines, the large number of seemingly arbitrary design decisions are not sufficiently explored. Instead, the paper reads mostly like a description of random components that happen to perform well in the particular benchmarks chosen. As a result, I believe the paper's contribution is quite limited and do not recommend acceptance.

--------

**Edit**

The authors have addressed several of my concerns in their replies and I have raised my score accordingly. Furthermore, it should be noted that my original statement that the authors do not provide source code was incorrect. I apologize for having overlooked it initially.

--------

**Edit 2**

I have re-reviewed the most recent revision of the paper. I would like to thank the authors for engaging with reviewers and for their updates to the paper. I believe the paper has improved as a result of the process, but will not be further raising my score at this time, and cannot argue strongly for acceptance of the paper.

While I believe there are interesting ideas presented in the paper, I continue to find the disconnect between motivation and experiments the most critical outstanding issue. I would encourage the authors to address this issue in future revisions of the work, as I feel if this issue was addressed, the work has the potential to be a significant contribution.

To clarify my meaning here, I will reiterate and elaborate on an example I used previously. In Section 3.2 the authors motivate the necessity of the cross-relational memory network component through the need to "reflect a holistic view of diverse preferences of users across different time slots" and then go on to give the following example "customers may first view some products in online retail sites, and make their purchase decisions one day after." While the ablation experiments show average performance increases when this component is included, they do not validate that this performance increase is due to the stated motivation. A more convincing experiment would identify examples of such behavior (a product view temporally followed by a purchase) and validate that including this component improves performance on these specific instances (and that performance degrades on these instances when the component is not included). Without this sort of analysis, it is impossible to disentangle the given motivation from other potential sources of performance improvement (regularizing effect, increased capacity, etc).

---

> ### Author Response · Authors · 2022-11-19
> **Details and Description of Sec.4.4 and Experiments(Part2)**
>
> * Weights visualization and EGCM modules：
>     * We have added the details of this module(*Appendix.A.9.2*). The following is a brief description of this experiment.
>     * We visualized the learned self-attention weight matrices $\Phi$ in Eq.4. Each weight matrix $\phi_{b,b'} \in \mathbb{R}^{|\mathcal{B}| \times |\mathcal{B}|}$ in $\Phi$ represents the relationship between the behaviors of a user. For example, for the Tmall dataset there are four behaviors *PageView, Favorite, Cart, Purchase*, therefore, $\phi \in \mathbb{R}^{4 \times 4}$, And, from the visualization results, it can be seen that the higher the number of interactions, the higher the weight of the behavior.
>     * The results of the experiment can show that our dynamic cross-relational memory network learns the semantic content of the data.
> * Embedding visualization and EGCM modules：
>     * We have added the details of this module(*Appendix.A.9.1*) of the supplementary material. The following is a brief description of this experiment.
>     * Visualizing behavior-specific embeddings in Fig.4 aims to show the influence of mutual information maximization on representation. Technically, we utilize t-SNE initialized with PCA[1]. And the experiment are conducted on datasets(Tmall, IJCAI) contain four types of behaviors(*page view/click, favorite, cart, purchase*). We can observe that the behaviors of EGCM are closer. In other words, for user $u$, embedding of other behaviors with the same index become closer, while users with different indexes $u\neq u'$ are pulled away.
>
> * Ablation Experiments of User Behavior:
>
>
> |         |        Tmall       |                    |                    |                  |   |        IJCAI       |                    |                    |                  |   |  E-commerce  |              |                  |
> |:-------:|:------------------:|:------------------:|:------------------:|:----------------:|---|:------------------:|:------------------:|:------------------:|:----------------:|---|:------------:|:------------:|:----------------:|
> |         | $w/o$-View | $w/o$-Fav. | $w/o$-Cart | Purchase |   | $w/o$-View | $w/o$-Fav. | $w/o$-Cart | Purchase |   | $w/o$-Review | $w/o$-Browse | Purchase |
> |  H@10  |       0.4625       |       0.5469       |       0.5338       |      0.3696      |   |       0.3546       |       0.4171       |       0.4634       |      0.3046      |   |    0.7323    |    0.7109    |      0.6768      |
> | N@10 |       0.2641       |       0.3265       |       0.3186       |      0.2295      |   |       0.1973       |       0.2341       |       0.2693       |      0.1773      |   |    0.4456    |    0.4412    |      0.4108      |
>
> * We appreciate the reviewers' thorough analysis. We have taken the insightful criticism and have rewritten(*Sec.4.4*) this module after reading the reviewers' remarks on the "in-depth examination of EGCM model" part. This, in our opinion, is a fascinating section of the article. It shows some visualization(self-attention weights, behavior-specific embedding, influence of $\tau$) results and combines the results with the theoretical analysis in the supplementary material(*Appendix.A.1*).
>
> * We appreciate the review's insightful remarks and provided references, which are necessary for our test procedure. In order to address potential issues with the test, we quickly implemented a version of the full-item test (without training data), chose baselines with superior results for comparison, and the experimental outcomes are as follows:
>
> |      |   Tmall  |          |   |   IJCAI  |          |   | E-commerce |          |
> |:----:|:--------:|:--------:|:-:|:--------:|:--------:|:-:|:----------:|:--------:|
> |      |   HR@10  |  NDCG@10 |   |   HR@10  |  NDCG@10 |   |    HR@10   |  NDCG@10 |
> | EHCF | 0.011751 | 0.005270 |   | 0.024629 | 0.012748 |   |  0.135780  | 0.065764 |
> |  CML | 0.013989 | 0.006279 |   | 0.029593 | 0.014911 |   |  0.139536  | 0.067804 |
> | EGCM | 0.015703 | 0.006932 |   | 0.035461 | 0.018640 |   |  0.151299  | 0.075318 |
>
>
> > [1]	Dunteman G H. Principal components analysis[M]. Sage, 1989.

---

> > ### Comment · Reviewer_Usaw · 2022-11-28
> > **non-sampled metrics, or full-item test results**
> >
> > This "full-item test" is appreciable. It is suggested to report the complete non-sampled metric results. In other words, similar to Table 2, it reports "Performance comparison of all methods on different datasets in terms of HR & NDCG" under the complete non-sampled metrics.

---

> > > ### Author Response · Authors · 2022-12-01
> > > **Further response on test sampling.**
> > >
> > >
> > > First off, we would like to express our gratitude to the reviewers for taking the time to read and comment favorably to our earlier response. We hope that some of your concerns regarding our work have been addressed by the earlier responses. **In fact, we are actively working on the experiments and will update them soon. Before obtaining valid experimental results, please allow us to elaborate on the issue.**
> > >
> > > * Reasons for sampling in previous works:
> > >
> > >     As GRU4Rec[1] states：
> > >     > "Recommender systems are especially useful when the number of items is large. Even for a mediumsized webshop this is in the range of tens of thousands, but on larger sites it is not rare to have hundreds of thousands of items or even a few millions. Calculating a score for each item in each step would make the algorithm scale with the product of the number of items and the number of events. This would be unusable in practice. Therefore we have to sample the output and only compute the score for a small subset of the items. "（GRU4Rec, ICLR2016）
> > >     > "
> > >
> > >     That is, the calculation of all item scores is more computationally intensive. For an online system with hundreds of millions of nodes, it is unreasonable not to sample. For this reason, test sampling is frequently utilized in academic research works. The point we are trying to make is that while test sampling is currently showing to be somewhat problematic, there are still legitimate reasons to use it.
> > >
> > >
> > > * Prior works using test sampling：
> > >     * > (Sec.5.1.2) Wang, Chenyang, et al. "Make it a chorus: knowledge-and time-aware item modeling for sequential recommendation." Proceedings of the 43rd International ACM SIGIR conference on research and development in Information Retrieval. 2020.
> > >     * > (Sec.4.2) Sun F, Liu J, Wu J, et al. BERT4Rec: Sequential recommendation with bidirectional encoder representations from transformer[C]//Proceedings of the 28th ACM international conference on information and knowledge management. 2019: 1441-1450.
> > >     * > (Sec.4.2) Li J, Wang Y, McAuley J. Time interval aware self-attention for sequential recommendation[C]//Proceedings of the 13th international conference on web search and data mining. 2020: 322-330.
> > >     * > (Sec.D) Kang W C, McAuley J. Self-attentive sequential recommendation[C]//2018 IEEE international conference on data mining (ICDM). IEEE, 2018: 197-206.
> > >     * > (Sec.4.1.2) Xia L, Xu Y, Huang C, et al. Graph meta network for multi-behavior recommendation[C]//Proceedings of the 44th International ACM SIGIR Conference on Research and Development in Information Retrieval. 2021: 757-766.
> > >
> > > > [1]Hidasi B, Karatzoglou A, Baltrunas L, et al. Session-based recommendations with recurrent neural networks[J]. arXiv preprint arXiv:1511.06939, 2015.

---

> > > ### Author Response · Authors · 2022-12-11
> > > **Updates the results of all item rank test experiments.**
> > >
> > > Dear reviewer, thank you for your previous responses. We have updated the results of all item rank test experiments in "Details and Description of Sec. 4.4 and Experiments (Part 2)" based on your insightful comments. And we made an effort to get as many results as we could in the limited time, and hope to address your concerns.
> > >
> > > |      |   Tmall  |          |   |   IJCAI  |          |   | E-commerce |          |
> > > |:----:|:--------:|:--------:|:-:|:--------:|:--------:|:-:|:----------:|:--------:|
> > > |      |   HR@10  |  NDCG@10 |   |   HR@10  |  NDCG@10 |   |    HR@10   |  NDCG@10 |
> > > | COTREC | 0.008818 | 0.003956 |   | 0.019122 | 0.009282 |   | 0.130600 | 0.064696 |
> > > | MBGMN  | 0.010908 | 0.004711 |   | 0.023311 | 0.010841 |   | 0.135366 | 0.068913 |
> > > | EHCF   | 0.011751 | 0.005270 |   | 0.024629 | 0.012748 |   | 0.135780 | 0.065764 |
> > > |  CML   | 0.013989 | 0.006279 |   | 0.029593 | 0.014911 |   | 0.139536 | 0.067804 |
> > > | EGCM   | 0.015703 | 0.006932 |   | 0.035461 | 0.018640 |   | 0.151299 | 0.075318 |
> > > | p-value | 1.6e-05 | 2.8e-06 |   | 2.4e-05 | 2.1e-05 |   |  7.7e-05  | 9.5e-05 |
> > >
> > > （*EHCF's framework is already all-item rank test, COTREC used all item's embedding matrix to calculate the score of allitem）

---

> > ### Comment · Reviewer_rXXA · 2022-12-02
> > **Reply to authors**
> >
> > I thank the authors for their thorough responses. I apologize for missing the included code earlier, it was included. Additionally, many of my concerns were addressed. I will be raising my score accordingly. My remaining concerns are detailed below.
> >
> > 1. My primary remaining concern, which was not addressed, is related to the lack of experimental evidence to support the stated motivation of some of the components. While I don't doubt that adding a particular component improves average performance (as demonstrated by the ablation studies), I believe the paper would be a more valuable contribution if the authors had designed experiments to demonstrate the components improve performance *because* of the specific motivation given. As an example, in Section 3.2 the authors motivate the necessity of the cross-relational memory network component through the need to "reflect a holistic view of diverse preferences of users across different time slots" and then go on to give the following example "customers may first view some products in online retail sites, and make their purchase decisions one day after." It seems it would have been easy to find examples of such behavior and validate that including this component improves performance there.
> > 2. I appreciate the authors providing results for non-sampled metrics in the response, but wonder why they weren't included in the revised paper or appendix. The numbers are also much lower in terms of absolute difference for sampled metrics, and p-values were not included, which makes me wonder if the differences in non-sampled metrics are statistically significant. The authors have mentioned in a comment elsewhere that ranking all items can be computationally difficult when there are many items, and sampled metrics are common. First, this should not be the case here, the largest dataset features 35K items. Second, we should adopt better methodology when it is available, not carry forward old methodology simply because it has been used in the past.
> > 3. There are some issues with the revised paper. For example, Table 2 is much less readable than it was in the original draft due to the folding, and Section 3.1.3 has a missing equation reference.

---

> > > ### Author Response · Authors · 2022-12-04
> > > **Response to the all item rank test experiment.**
> > >
> > >
> > > * The experiments for the all item rank test were placed in Supplementary Material A.8.
> > > (Due to the lack of time left in the previous dealine they were not described in detail.)
> > > * Without sampling, the number of competitors involved in the ranking on the three datasets will be 315 times, 359 times, and 18 times, respectively, when calculating the metrics. Therefore, it is reasonable to have a certain range of reduction in each indicator. We were honest in our recording of the experiments' outcomes.
> > > * The suggestion to add p-values is reasonable, and we sincerely thank you for pointing out this. We will add the corresponding experimental results in our revision.

---

> > > ### Author Response · Authors · 2022-12-04
> > > **Details of experiments that can explain the effects of the developed components.**
> > >
> > > First of all, your prompt reply to our response has our sincere gratitude. And we also appreciate your follow-up feedback regarding our work. Please allow us to respond to your questions.
> > >
> > > (In fact, in previous response "Details and Description of Sec. 4.4 and Experiments (Part 1)", the function of the designed modules has been briefly mentioned.)
> > > * Weight Visualization Experiment(the example mentioned in the reviewer's reply): As our response "Differences from Previous Work(Part2)" shows，to learn the relationship between multiple behaviors of users across time slot, we visualize the learned self-attention weight matrices $\phi \in \mathbb{R}^{|\mathcal{B}| \times |\mathcal{B}|}$ in figure 2(f) for 400 sampled users(with Page View, Favorite, Cart, Purchase four kinds behaviors). Here, $\phi = q_t \cdot k_{t-1}^T$, and $q_t, k_{t-1}^T$ are multi-behavior information for user at time $t, t-1$, respectively. Thus, the self-attention weight matrix shows the relationship of each behavior to each other. In particular, the behavior is from the adjacency time slot. Then, we select six cases from and combine the behavior-specific interaction number to analyze the the learned weight matrices. It can be observed in Figure 2(f) that most matrices have the darkest diagonal color, which is the characteristic of self-attention. Additionally, the weight's value increases with the amount of behaviors. For example, user 22186 with 111 interactions in page view behavior, the number of interactions in other behaviors are 1, 5, 4 which are too small relatively. Thus it is difficult to learn differentiated values in the other three rows relative to the first row of the weight matrix.
> > > * Ablation Experiment: If the GRU($r/w-$GRU) is used instead of the memory module(in Eq.4,5) to delivering information across time slots, the result will be suboptimal. (For explanations of other experiments, see previous response "Details and Description of Sec.4.4 and Experiments(Part1)")

---

> ### Author Response · Authors · 2022-11-19
> **Details and Description of Sec.4.4 and Experiments(Part1)**
>
> For the modules designed in the EGCM framework, in addition to the main experiments(*Tab.2*) that can illustrate the overall results, the ablation experiments(*Tab.3*), parameter experiment(*Fig.2(e)*), the visualization experiments of weights(*Fig.2(f)*) and embedding(*Fig.2(a)-(d)*) can verify the validity of the corresponding modules. Moreover, we add experiments on ablation behavior(Appendix.A.8) to illustrate the contribution of behavioral information to the results. In the following, we describe each module in detail. (We released the codes and the reviewer may not have noticed it before.)
>
>
> * Ablation experiments and EGCM modules：
>     * Ablation experiments can demonstrate the effectiveness of the main modules of our model：
>         * w/o-CL: The result of remove contrastive based on w/o-JBL. And it can be seen that especially on Tmall and IJCAI, contrastive learning contributes a lot to the results. The reason may be that these two datasets have more behavioral data and larger overlap of timestamps.
>         * r/w-GRU: The result that to remove the self-attention based memory module and model the sequential relationship between time-slots with GRU as a replacement.
>         * w/o-MBG: The result of remove the short-term multi-behavior graph encoder based on w/o-JBL.
>     * The main contributions to the superior results of our EGCM are the introducing of behavior-aware graph neural encoder(w/o-MBG), the use of multi-behavior data(w/o-JBL, the supplemented the multi-behavioral ablation experiment(*Appendix.A.8*) and contrastive learning(w/o-CL).

---

> ### Author Response · Authors · 2022-11-19
> **Model Design and Detailing Decisions**
>
> Experimental efforts determine each module of EGCM. Before being incorporated into the model architecture, every module has undergone meticulous design and testing. For example:
> * The representation learning for user and item in Eq.1, 2 can be verified using ablation experiments w/o-MBG to verify the effect.
> * The normalization of the adjacency matrix in Equation 2 is different from the version mentioned in the GCN paper[1] (the normalized Laplacian of GCN is symmetric) and is adapted to the user-item bipartite graph, which is a better choice for experimental results(*Details are placed in Appendix.A.5*).
> * The initialization of time information transfer in Eq.3 is also the result of several versions of the experiment. Alternative versions are concatenate, sum, mean or without the combine.
> * Eq.4,5 refer to the dynamic cross-relational memory network(Sec.3.2) to model multiple behavioral relationships across time-slots. The messaging across time slots can be replaced by GRU, i.e. ablation experiments r/w-GRU(Sec.4.3). The results prove the effectiveness of the module we designed.
> * The attention in Eq.5 is still the result after trying sum, mean, concatenate+transformation.
> * InfoNCE[2] in Eq.7 is still the result after multiple attempts. In the denominator, the second term in $\sum$ is because of the better results after adding the negative samples which are in the same view of the anchor node.
> * LightGCN[3] is a very well-known work in recommender systems. We have also done related experiments hoping to use a lightweight GNN architecture. However, the results of the experiments support us to use the current implementation in the model framework. We think that the relevant reason may be the complex relationship of multi-behavior data, and the use of linear layers and activation functions helps to fit.
>
> > [1]Kipf T N, Welling M. Semi-supervised classification with graph convolutional networks[J]. arXiv preprint arXiv:1609.02907, 2016.
> > [2]Oord A, Li Y, Vinyals O. Representation learning with contrastive predictive coding[J]. arXiv preprint arXiv:1807.03748, 2018.
> > [3]He, Xiangnan, et al. "Lightgcn: Simplifying and powering graph convolution network for recommendation." Proceedings of the 43rd International ACM SIGIR conference on research and development in Information Retrieval. 2020.

---

> ### Author Response · Authors · 2022-11-19
> **Further clarification of model technical novelty and design motivation**
>
> Thank you so much for your provided constructive and helpful comments. Please find our detailed response and clarifications below. If our responses resolve your questions, we really genuinely hope your consideration in raising your rating score.
>
>
> At the core of our dynamic multi-behavior recommender system is to effectively model the dynamic interaction heterogeneity and the implicit cross-type behavior dependencies. However, this is not trivial and remain unexplored in existing works due to the following techniques challenges:
>
> * Dynamic interaction heterogeneity of sequential multi-behavior patterns: To tackle this challenge, we design a novel dynamic cross-relational memory network to distill the time-evolving item-item relationships at the fine-grained level of user preferences. In our new dynamic cross-relational memory module, both short-term and long-term multi-behavior intents of users are characterized within the dynamic behavior heterogeneity context, which has not been studied in current works.
> * Jointly modelling of multi-behavior commonality and user preference diversity: To tackle this challenge, we innovatively integrate the contrastive learning with dynamic multi-behavior modelling. We propose a multi-behavior contrastive learning paradigm to supercharge our recommender to preserve multi-behavior commonality and user-specific interest diversity. In particular, we generate our contrastive representation views with the type-specific behavior semantics and the aggregated user multi-behavior embedding. To simultaneously encode user multi-behavior commonality and interests diversity, our EGCM enables the behavior-level contrastive augmentation with positive pairs by pulling the type-specific behavior representation and multi-behavior embedding of the same user. Behavior embeddings of different users are regarded as negative samples are pushed away to capture diverse user behavior-aware preference.

---

### Official Review · Reviewer_Usaw · 2022-10-28

**Confidence:** 5
**Clarity, Quality, Novelty And Reproducibility:** writing is good; technical originalit…
**Correctness:** 3
**Technical Novelty And Significance:** 2
**Empirical Novelty And Significance:** 3
**Recommendation:** 6

**Strength And Weaknesses:**

Strong & Weak:

S1: It seems novel to use contrastive learning framework for the sequential & multi-behavior recommender systems. Although both sequential and multi-behavior ResSys methods are proposed in existing works, this paper is the first to adopt contrastive learning technique to construct the two auxiliary tasks to better learn the main task. Furthermore, the ablation study shows that, the proposed model degrades the performance by a large margin if the contrastive learning component is removed, especially on the Tmall and IJCAI datasets, though the impact of contrastive learning is not so big on the E-commerce dataset.

W1: One possible weakness in my own opinion: While the proposed EGCM is a new method for sequential & multi-behavior recommender systems, each of the individual piece of EGCM (multi-relation GNN, Transformer, contrastive learning) is not new in itself and has been widely adopted in existing works as pointed out in the related works. This may be not a big issue in practice, since EGCM demonstrated a strong empirical result on three datasets.


**Summary Of The Paper:**

Summary:

This paper proposed to care about the sequential multi-behavior recommender systems where the multiple behaviors include a subset from {view, favorite, cart, purchase, browse, review}. To capture the multiple user behaviors, a multi-relation deep GNN is adopted to propagate behavior-aware messages and the graph Laplacian normalized function is incorporated into the message passing. In this way, both item representations and user’s short-term interests can be captured. As for the relations among multiple behaviors, The paper is to learn cross-type behavior dependency matrix by a Transformer, or called self-attention-based memory network in the paper. In this way, both sequential/time-evolving and multi-type behavior dependencies are learned. In the end, the paper proposes a contrastive learning to pull the type-specific behavior and multi-behavior representation of the same user to be closer as positive pairs while to push the behavior of different users away as negative pairs. Experiments are conducted on three datasets by comparing with various kinds of 19 baselines. Ablation and visualization are further shown to understand the components of the proposed EGCM model.


**Summary Of The Review:**

Comments:

C1: some typos. On page 2, “an” in the statement “we propose a Evolving Graph”. On page 6, the “+” symbol in computing $\tilde E_{t,u}$; the subscript $e_i$ should be $e_u$ in the statement “by pulling the type-specific behavior embedding…”; the dot product in the statement “measured by the doc product…”

---

> ### Author Response · Authors · 2022-11-19
> **Statement of the Innovation and Motivation of EGCM**
>
> First of all, we appreciate the reviewer for raising this question and please allow us to give explanation with the following details. In response to your comments, we have made modification in our revised paper to further clarify the difference and advantage of our newly proposed method over existing techniques.
>
> Each module of our EGCM framework is specially designed to tackle the unique challenge of the multi-behavior sequential recommendation task. While some existing techniques (e.g., graph neural networks, contrastive learning, transformer) are adopted as the backbone or learning paradigm in our method, how to explicitly capture the behavior heterogeneity in a dynamic environment with evolving user preference, requires non-trivial tailored modelling. The difference between our EGCM and existing techniques are elaborated from the following three perspectives:
> * Transformer. Different from leveraging Transformer as the single sequence encoder in most existing Transformer methods, our multi-behavior sequential recommender system involve heterogenous behavior sequences and their implicit cross-behavior dependencies, which cannot be easily handled by existing Transformer solutions. To capture the time-evolving dependencies across different types of behaviors, we design a Transformer-like dynamic memory network(*Sec.3.2*) to take the type-aware behavior embeddings from temporally adjacent time slots as input, and then calculate their correlation weights based on self-attentive scheme. With our new tailored Transformer-based memory network, the dynamic behavior heterogeneity can be well preserved in our encoded time-aware behavior representations.
>
> * Contrastive Learning. Following the self-supervised learning (SSL) line, contrastive learning can be considered as a general learning paradigm which generates auxiliary self-supervision signals through instance contrasting. Inspired by the contrastive learning, we design new contrastive learning framework(*Sec.3.3*) to fit the dynamic heterogenous interaction modelling, with the aim of preserving both the multi-behavior commonality and user-specific preference diversity.
>
>     * Commonality(*Theoretical Analysis: Appendix.A.1.1*): We believe that there are user common preferences among multiple behaviors. By performing mutual information maximization across behaviors, more accurate and robust user representations can be obtained. The theoretical analysis section also states that minimizing our multi-behavioral cl loss is approximately a lower bound for maximizing multi-behavioral mutual information(*A.1.1.Eq.14*). In addition, the theoretical analysis section provides us with a theoretical analysis for multi-view contrastive learning(*A.1.1.Eq.15*).
>     * Diversity(*Theoretical Analysis: Appendix.A.1.2*): The modeling of short-term user-item interactions is based on GNN, which helps to model higher-order connectivity[1]. However, the multilayer GNN architecture introduces over-smoothing problems. The theoretical analysis part points out that the cl loss gives a larger gradient to the difficult negative samples(*A.1.2.Eq.21*), which means push away similar user representations, i.e., mitigating the over-smoothing problem. our experiments(*Sec.4.4 and A.9.3*) also corroborate with the theoretical analysis. That is, a small temperature coefficient does give a larger gradient, but too large may lead to a gradient explosion.
>
> * Graph Neural Network. Our proposed multi-behavior graph neural network has the following advantages as a short-term multi-behavior encoder.
>     * The main point is that GNN is divided into time slots(*Sec.3.1.1*), and there is a part that passes information between time slots. And our self-attention is specifically used to model the multi-relationship between time slots(*Sec.3.2*).
>     * GNN's initialization module(*Sec.3.1.3*) takes over the information from the previous time-slot.
>     * Encodes heterogeneity for multi-behavior.
>     * Encodes the information of short-term.
>
>
> > [1]Wang X, He X, Wang M, et al. Neural graph collaborative filtering[C]//Proceedings of the 42nd international ACM SIGIR conference on Research and development in Information Retrieval. 2019: 165-174.

---

> ### Author Response · Authors · 2022-11-19
> **Typo Fixing**
>
> Thanks to reviewer for pointing out the typos in our paper. We have fixed the syntax error in our revision(*Sec.3.4*). We also proofread our paper to eliminate identified typos.

---

### Author Response · Authors · 2022-11-19
**General Response**

We appreciate the constructive criticism and thoughtful suggestions from every reviewer. Thank you very much for your thoughtful consideration.

We are pleased to learn that the reviewers think our work is "novel and interesting", and "solves an under-explored topic" [Reviewer HypT], and that the paper's "overall quality looks good" [Reviewer Usaw]. We also genuinely appreciate the valuable suggestions from the reviewers, such as changing the technique section [Reviewer Usaw, Reviewer 1ytF], changing the sampling in the test section to all item rank, and adding tests to demonstrate the usefulness of the model module [Reviewer rXXA].

Below, we give each reviewer a lengthy response that is precise. We also modified our manuscript in response to the reviewer's criticisms. The significant revisions are outlined, as well as  the significant modifications are listed.

METHODOLOGY: To make this module more concise and easy to follow, we reduced unnecessary formulas, modified notation, and reorganized the structure.(suggested by Reviewer Usaw, 1ytF)

4.4 In-depth Analysis Of Egcm Model: To further explore the effects of each module, explore explainability, and establish links with theoretical analysis, we have added more details.(suggested by rXXA)

APPENDIX.7: To illustrate the differences between the modules in Sec. 3.2 and previous work, we present a table of dimensional transformations. (response to 1ytf).

APPENDIX.8(1): To account for the factors contributing to the model results, we conducted ablation experiments of the behavior.(suggested by rXXA)

APPENDIX.8(2): To eliminate the effect of test sampling, we conducted experiments with all item rank tests and compared other baselines with the same settings(suggested by rXXA).

APPENDIX.9: To give the reader a clearer picture of the effect of our module and the experiments corresponding to it, we have expanded the details of experiments.

---

### Decision · Program_Chairs · 2023-01-20

**Decision:**

Reject

**Justification For Why Not Higher Score:**

The scores the paper currently receives (10, 6, 6, 6) certainly don't suggest a clear rejection. There have been quite a few score updates during the discussion phase where initially two reviewers gave rejections but later changed to marginally accept. However, during the discussion, one reviewer who gave it a 6 still feels that this paper is not good enough to be accepted, which makes me recalibrate the scores. (I myself also read the paper and I think I would give it a 4 or 5 if I were a reviewer.) Furthermore, I don't find the comments from reviewer HypT who gave this paper a 10 very helpful -- this paper clearly has some problems (as pointed out by other reviewers), yet despite us asking for more input, reviewer HypT did not participate in any of the discussions. Therefore, his/her comments did not get much weight in my decision.

**Justification For Why Not Lower Score:**

N/A

**Metareview: Summary, Strengths And Weaknesses:**

Summary: This paper presents Evolving Graph Contrastive Memory Network (EGCM) which aims to incorporate multiple types of behavior signals (click, purchase, review, etc.) in a sequential recommendation setting (i.e., learning the ever-evolving user preference over time). There are a few components in EGCM. First, a graph neural network encoder is used to capture the short-term preference with different behavior types.  Following that, a transformer-style attention mechanism is used to capture the long-term multi-behavior preference from the short-term representations as a memory network. Finally, a contrastive learning objective is adopted to make sure the behavior-specific representation and behavior-aggregated representation from the same user stay close (positive pairs) while pushing the representations from different users further away (negative pairs). Experimental results demonstrate that the proposed EGCM outperforms other baselines. Ablation studies are also conducted to demonstrate the importance of each individual component.

Strengths: Extensive experimental results show that the proposed EGCM works well over other baselines. The authors did make an effort at demonstrating the effect of various components via ablation studies.

Weaknesses: There have been quite a few score updates during the discussion phase. I have also read the paper myself in the past week and discussed with some of the reviewers regarding their opinions about the paper. Some of the minor weaknesses are mostly addressed. To me there still seems to be a major weakness:

* The paper right now reads overly complicated with limited technical novelty (Reviewer rXXA and 1ytF). As the summary above shows, it consists of a few components, each of which involves some modifications to an existing piece of work. Many of the design choices seem rather arbitrary and are not adequately justified -- I am sure these choices all helped but as with many developments with improving the architecture of a neural net, we are practically faced with infinitely many choices. I don't doubt that adding a particular component improves the performance, especially given the relatively smaller datasets used in the experiments (more regularization will help here). But the degree of specificities in this paper at the moment seems too much. It almost reads like a combination of specially-designed components which are put together to do well on the benchmark datasets. If we are one day faced with a similar multi-behavior dynamic recommendation problem, I am not sure how much learning we can get from this paper. (It might not be the most accurate analogy, but this almost feels like the winning solution of a competition which is too complicated that people will end up using a simplified version in practice.)
* As for in which direction the authors can improvement the paper, I think adequately justifying many of the design decisions is in order (maybe even simplifying things a bit). To quote a reviewer during the discussion phase: "I believe it would be much more valuable if the authors had designed experiments to demonstrate the components improve performance because of the specific motivation given. As an example, in Section 3.2 the authors motivate the necessity of the cross-relational memory network component through the need to "reflect a holistic view of diverse preferences of users across different time slots" and then go on to give the following example "customers may first view some products in online retail sites, and make their purchase decisions one day after." It seems it would have been easy to find examples of such behavior and validate that including this component improves performance there."
* There are also some concerns around the sampled metrics vs full-catalog metrics. Given the choice the datasets, I don't think it is particularly challenging to compute the full-catalog metrics as the authors claimed. Update: I noticed that the authors just provided the full-catalog metrics. And as some of the reviewers suspected they are indeed much closer than the sampled metrics, which begs the question: do we really need so many over-complicated components to be able to achieve practically (not statistically, since none of the error bars in the experiments actually means much) significant results?

I also think this paper will find a larger audience in a more recommender-systems-focused venue (e.g., KDD, TheWebConf, etc.)